# Evaluation of pragmatic oxygenation measurement as a proxy for Covid-19 severity

Maaike C. Swets [1,2,72], Steven Kerr[1,3,72], James Scott-Brown [4], Adam B. Brown [1], Rishi Gupta [5], Jonathan E. Millar[1], Enti Spata[6], Fiona McCurrach[7], Andrew D. Bretherick [8], Annemarie Docherty [3], David Harrison[9], Kathy Rowan [9], Neil Young[10], ISARIC4C Investigators*, Geert H. Groeneveld[2], Jake Dunning[11], Jonathan S. Nguyen-Van-Tam [12], Peter Openshaw [13], Peter W. Horby [11], Ewen Harrison[3], Natalie Staplin[6], Malcolm G. Semple [14,15], Nazir Lone [3,16] & J. Kenneth Baillie [1,16,17,18] ✉

Choosing optimal outcome measures maximizes statistical power, accelerates discovery and improves reliability in early-phase trials. We devised and evaluated a modification to a pragmatic measure of oxygenation function, the $S/F$ ratio. Because of the ceiling effect in oxyhaemoglobin saturation, $S/F$ ratio ceases to reflect pulmonary oxygenation function at high $S_pO_2$ values. We found that the correlation of $S/F$ with the reference standard ($P_aO_2/F_IO_2$ ratio) improves substantially when excluding $S_pO_2>0.94$ and refer to this measure as $S/F_{94}$. Using observational data from 39,765 hospitalised COVID-19 patients, we demonstrate that $S/F_{94}$ is predictive of mortality, and compare the sample sizes required for trials using four different outcome measures. We show that a significant difference in outcome could be detected with the smallest sample size using $S/F_{94}$. We demonstrate that $S/F_{94}$ is an effective intermediate outcome measure in COVID-19. It is a non-invasive measurement, representative of disease severity and provides greater statistical power.

Therapeutic research in COVID-19 depends on efficient, accurate assessment of therapeutic candidates in early-stage clinical studies. Efficacy measures should be "clinically meaningful"[1] endpoints, such as the WHO ordinal scale[2]. Intermediate endpoints for early phase trials, or severity measures for observational studies, must be modifiable by therapy and ideally should have a continuous numerical distribution to improve statistical power[3]. The endpoint should accurately predict the definitive outcome of interest and

[1]Roslin Institute, University of Edinburgh, Edinburgh, UK. [2]Department of Infectious Diseases, Leiden University Medical Center, Leiden University, Leiden, The Netherlands. [3]Centre for Medical Informatics, Usher Institute, University of Edinburgh, Edinburgh, UK. [4]School of Informatics, University of Edinburgh, Edinburgh, UK. [5]Institute for Global Health, University College London, London, UK. [6]Medical Research Council Population Health Research Unit at the University of Oxford, Nuffield Department of Population Health (NDPH), Oxford, UK. [7]EMERGE, NHS Lothian, Royal Infirmary Edinburgh, Edinburgh, UK. [8]MRC Human Genetics Unit, Institute of Genetics and Cancer, University of Edinburgh, Western General Hospital, Edinburgh, UK. [9]Intensive Care National Audit & Research Centre, London, UK. [10]Department of Anaesthesia, Critical Care and Pain Medicine, Royal Infirmary of Edinburgh, Edinburgh, UK. [11]Pandemic Sciences Institute, University of Oxford, Oxford, UK. [12]Population and Lifespan Health, University of Nottingham School of Medicine, Nottingham, UK. [13]National Heart and Lung Institute, Imperial College London, London, UK. [14]Institute of Infection, Veterinary and Ecological Sciences, Faculty of Health and Life Sciences, University of Liverpool, Liverpool, UK. [15]Department of Respiratory Medicine, Alder Hey Children's Hospital, Liverpool, UK. [16]Intensive Care Unit, Royal Infirmary of Edinburgh, Little France Crescent, Edinburgh, UK. [17]Baillie Gifford Pandemic Science Hub, Centre for Inflammation Research, University of Edinburgh, Edinburgh, UK. [18]MRC Human Genetics Unit, Institute for Genetics and Molecular Medicine, University of Edinburgh, Edinburgh, UK. [72]These authors contributed equally: Maaike C. Swets, Steven Kerr. *A list of authors and their affiliations appears at the end of the paper.
✉e-mail: j.k.baillie@ed.ac.uk

ideally should also be closely related to the causal pathway to this outcome.

In COVID-19, efficacy measures such as the WHO ordinal scale, duration of hospitalisation, and viral load have been used widely[4,5]. Both the WHO ordinal scale and various alternative ordinal scales[6,7], rely on a complex clinical measure - the level of respiratory support received by a patient - as an indicator of illness severity. Viral load is a valid outcome for antiviral therapy, but it has not been shown to correlate with mortality benefit, and is not directly relevant to the effect of anti-inflammatory treatments[8–10]. In the RECOVERY trial, we identified a need for more powerful intermediate endpoints for early-phase clinical trials.

Impairment of the pulmonary oxygenation function indicates disease progression in COVID-19[11], and is strongly predictive of mortality[12]. Importantly, in COVID-19, failure of pulmonary oxygenation is likely to be mechanistically linked to death: patients at extreme risk of mortality[12] have high survival rates if oxygenation is provided by extracorporeal membrane oxygenation (ECMO)[13]. Pulmonary oxygenation function, together with clinical decision-making and resource availability, determines movement between most of the stages of the WHO Ordinal Scale (WHO scale points 4-9)[2]. Oxygenation function is a key determinant of efficacy for immunosuppression with corticosteroids in COVID-19[9]. It is likely that pulmonary oxygenation function lies on the causal pathway between the SARS-CoV-2 infection and death for many hospitalised patients.

Peripheral oxygen saturation can be measured easily and non-invasively using a pulse oximeter (formally, arterial oxygen saturation measured by pulse oximetry, rather than direct measurement in blood, is $S_pO_2$). The ratio of $S_aO_2$ or $S_pO_2$ to inspired fraction of oxygen ($F_IO_2$), known as the $S/F$ ratio, provides a continuous index of pulmonary oxygenation function which can be calculated without an arterial blood sample. $S/F$ correlates well with the most widely-used arterial blood-derived measure of oxygenation - $P/F$ ratio ($P_aO_2/F_IO_2$)[14]. $S/F$ under steady state conditions in humans can range from around 0.5 (severe oxygenation defect) to 4.8 (perfect oxygenation function). A major limitation of $S/F$ is the ceiling effect: at high $S_aO_2$ values, $S_aO_2$ ceases to be dependent on pulmonary oxygenation function, because the blood is close to maximally oxygenated and the relationship between the P/F ratio and the S/F ratio is non-linear[15,16]. For example, a healthy patient with perfect lungs breathing 21% oxygen with $S_aO_2 = 0.99$ would have $S/F = 4.7$, but the same patient breathing 100% oxygen would have $S/F = 0.99$.

In order to improve the accuracy of measurement of lung oxygenation, we propose limiting the ceiling effect in prospective data by protocolising measurement of $S_pO_2$ to control high values or in retrospective (opportunistic) analyses by excluding values recorded with $S_pO_2$ above a given threshold value. We first evaluated an optimal threshold using both synthetic and real data from arterial blood gas (ABG) samples, predicting that $S_pO_2 \leq 0.94$ provides optimal predictive validity, at a level of induced hypoxia that is broadly acceptable to clinicians.

We defined the $S/F_{94}$ measurement as $S/F$ measured when $S_aO_2 \leq 0.94$ or $F_IO_2 = 0.21$. In opportunistic data, $S/F_{94}$ can be estimated by excluding $S_pO_2$ values above 0.94 unless $F_IO_2 = 0.21$. In prospective, protocolised measurements, $S_aO_2 \leq 0.94$ can be achieved by reducing $F_IO_2$ to a minimum of $= 0.21$ (the fraction of oxygen in ambient air). Since many patients receive oxygen through devices for which $F_IO_2$ is not accurately quantified (e.g. Hudson mask, nasal cannulae), prospective studies measuring $S/F_{94}$ will require a modification of oxygen delivery devices which, in itself, is expected to improve the accuracy of measurement (Appendix: Protocol).

In order to assess $S/F_{94}$ as an outcome measure, we first used physiological model to evaluate the relationship with a reference standard, the $P/F$ ratio. Second, we compared the predictive validity of $S/F_{94}$ with several other measures of pulmonary oxygenation function,

including the $S/F$ ratio and the alveolar-arterial difference (A-a). We then used the ISARIC4C dataset to train models for a range of intermediate outcomes, including the WHO ordinal scale and $S/F_{94}$, as predictors of 28-day mortality. We used these models to estimate sample sizes that would be required to see a given treatment effect. Finally, using data from the RECOVERY trial we estimated the expected improvement in the required sample size when using a protocolised, rather than opportunistic, $S/F_{94}$ measurement.

## Results
### Relationship with the reference standard oxygenation measure ($P/F$)
There is a consistent pattern in both synthetic (Fig. 1) and real (Supplementary Fig. 1) data: if no maximum cut-off value for $S_aO_2$ is used, spuriously low $S/F$ values are seen in patients with good lung function, reflected in high $P/F$ values (Fig. 1a, Supplementary Fig. 1a). This is due to the ceiling effect - $S_aO_2$ cannot rise above 100%. These misleading values are removed by excluding values with $S_aO_2$ above 94% (Fig. 1b, Supplementary Fig. 1b), which improves the correlation with the reference standard for both synthetic (Spearman $S/F$: 0.40; $S/F_{94}$: 0.85; Fig. 1d) and real data (Spearman $r$ $S/F$: 0.82; $S/F_{94}$: 0.97, Supplementary Fig. 1c).

### Predictive validity
In parallel, we assessed the predictive validity of $S/F$ and $S/F_{94}$. As in our previous work[17], we assert that if $S/F_{94}$ is measuring true oxygenation function well, then it should be able to more accurately predict a future event: the $P_aO_2$ value in a future arterial blood gas measurement taken from the same patient. We used a pre-existing dataset of unselected ABG result pairs from hospitalised patients, described in detail previously[17]. We quantified the MAE above baseline in $P_aO_2$ to quantify predictive validity, with lower error values indicating better performance (Fig. 1c, Supplementary Fig. 2). Across a range of maximum cut-off values for $S_aO_2$, the lowest MAE value was obtained at 94% (Fig. 1c; $S/F$ MAE = 4.41 kPa (IQR: 2.74-6.63 kPa); $S/F_{94}$ MAE = 3.32 kPa (IQR: 1.87-5.26 kPa), p(MWU) = $3.7 \times 10^{-18}$).

### Evaluation in ISARIC4C data
39,765 cases in the ISARIC4C study had $S_pO_2$, $F_IO_2$ and clinical data available for analysis and met the inclusion criteria (see Methods). Mortality in this population was 20.8% (Table 1). Since measurement of $S/F_{94}$ was not protocolised in ISARIC4C, measurements were obtained for patients for whom $S_pO_2$ happened to be $\leq 0.94$ or who were breathing room air ($F_IO_2 = 0.21$), therefore meeting the $S/F_{94}$ definition. The conceptual advantage of $S/F_{94}$ over $S/F$ is that it offers a closer relationship to the pathophysiological process of interest. This is not expected to be apparent in the distribution of values observed, but rather in the sensitive detection of a real therapeutic effect. For this reason, and because of the risk of selection bias (see Methods), we did not undertake a direct comparison of patients meeting the criteria for $S/F_{94}$ measurement, against patients who do not. Instead, we evaluated $S/F_{94}$ against other commonly used outcome measures.

In order to select the timepoint of $S/F_{94}$, several aspects were taken into account. Firstly, we looked at data availability. Within the ISARIC4C dataset, S/F values were available for the largest numbers of patients on days 0, 2, 5 and 8 from study enrolment. Second, among patients who remained in hospital, the distribution of $S/F_{94}$ values moves over the first few days from study enrolment towards a bimodal pattern with high values in survivors, and low values in non-survivors (Fig. 2a). Finally, in order to make a meaningful comparison with the $S/F_{94}$ at the day of enrolment, we preferred timepoints that were at least a few days after enrolment. We therefore chose day 5 as the primary timepoint for comparison. The distribution of measured $S/F_{94}$ values and assigned maximum/ minimum values for those who were discharged/ died can be seen in (Fig. 2b). On day 5, 1077 out of 7,312

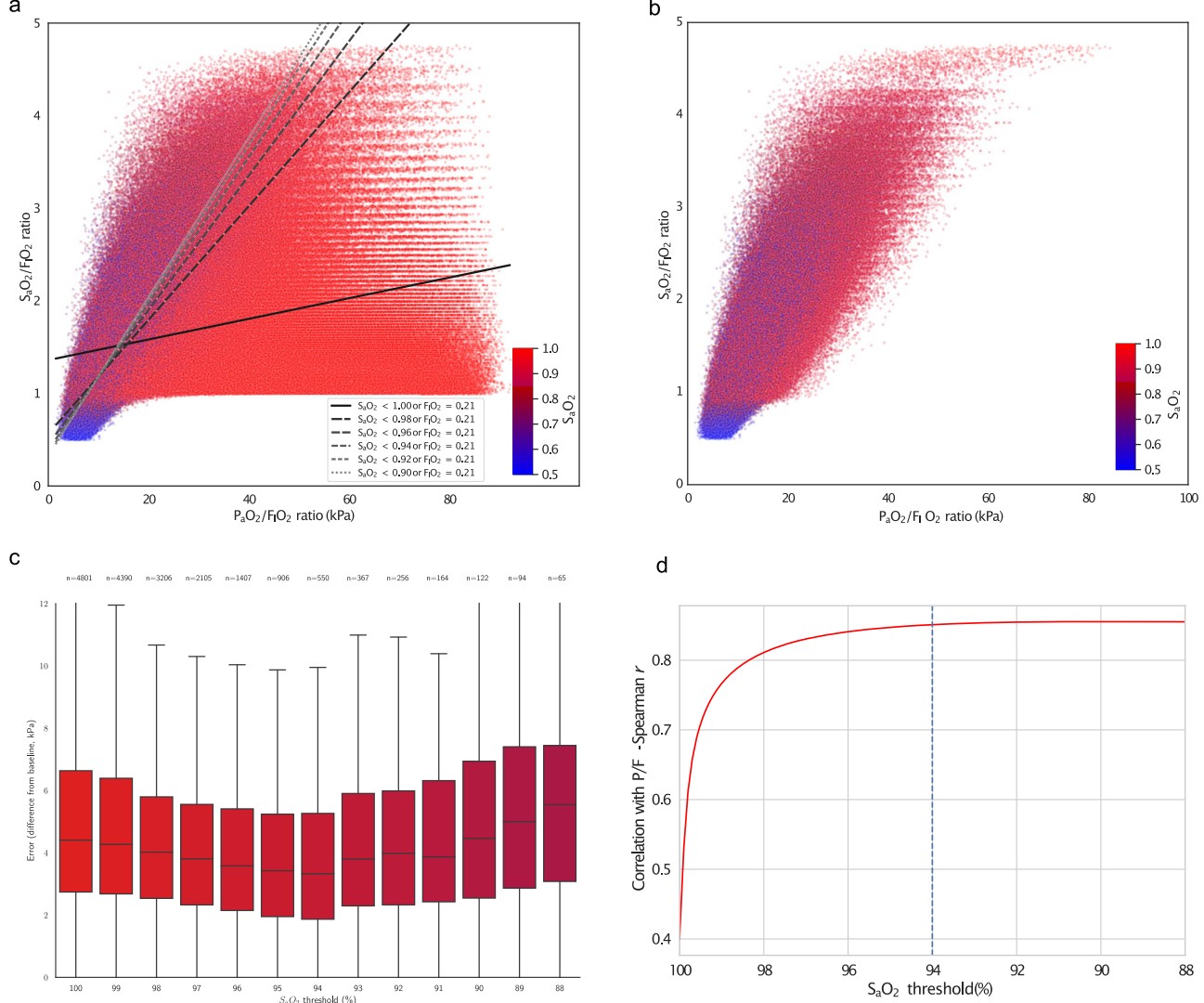

**Fig. 1 | Comparison of $P/F$ and $S/F$ or $S/F_{94}$ in synthetic data. a, b** Scatterplots of $P/F$ vs $S/F$ individual measurements across a range of hypothetical physiological characteristics. Points are coloured according the $S_aO_2$ as shown in the colour scale. (**a**) including all values, showing linear regression of $S/F$ against $P/F$ in using different cut-off values for $S_aO_2$. Patients breathing air ($F_IO_2 = 21\%$) were included in all bins. (**b**) including only values with $S_pO_2 \leq 0.94$ or $F_IO_2 = 21\%$ (**c**) Optimisation of cut-off value for $S_aO_2$ using predictive validity: the error in the prediction of a future $P_aO_2$, based on a previous one (using a pre-existing dataset of ABG results[17]). Centre line represents median values, box limits represent upper and lower quartiles, whiskers represent minimum and maximum values. (**d**) change in correlation coefficient (Pearson's $\rho$) as the threshold for inclusion is lowered from $S_aO_2 < 100\%$ to $S_aO_2 < 80\%$.

(14.7%) known $S/F_{94}$ values were an assigned maximum/minimum value due to death/discharge. On day 8, 1948 out of 6079 (32.0%) known $S/F_{94}$ values were an assigned maximum/minimum value. A sensitivity analysis excluding these assigned values is in the supplementary material.

An intermediate clinical outcome should have a strong association with a definitive outcome. Using 28-day mortality as the definitive outcome, and including $S/F_{94}$ values on both day 0 and day 5 as covariates in a linear regression model, we found a strong inverse association between $S/F_{94}$ on day 5 and mortality: an increased risk of mortality at day 28 is associated with a lower value of $S/F_{94}$ on day 5 (Fig. 2d). The OR for 28-day mortality is 0.25 (95% confidence interval 0.23-0.28), meaning that for a 1 unit increase in $S/F_{94}$ on day 5, the odds of 28-day mortality decrease by 75%.

We also compared $S/F_{94}$ with a widely used intermediate outcome, the WHO scale. Since this scale records clinical decisions about therapy that are, in part, determined by the severity of hypoxic lung disease, a close relationship was expected with $S/F_{94}$ (Fig. 2c). The

distributions were consistent between patients meeting the inclusion criteria (Fig. 2c) and unselected patients (Supplementary Fig. 5a). The distribution of $S/F_{94}$ values between outcomes at day 28 for patients meeting the inclusion criteria is similar on day 0 and day 5 (Supplementary Fig. 5b and Supplementary Fig. 5c). As expected, when there are no criteria for supplemental oxygen in the first 3 days since admission (unselected patients, Supplementary Fig. 5d and Supplementary Fig. 5e), there is a relative increase of patients with high $S/F_{94}$ values on day 0.

**Sample size estimation**

Using the observed relationships in ISARIC4C data for eligible patients (see Methods), we quantified effect sizes associated with a 15% relative risk reduction in mortality for each of the following measures: $S/F_{94}$ at 5 and 8 days after study enrolment, the WHO ordinal scale at 5 and 8 days after study enrolment, the proportion of patients who reached a sustained 1 or 2-level improvement on the WHO ordinal scale, and a definitive outcome, 28-day mortality. We chose a 15% relative risk

**Table 1 | Comparison of outcome measures among 39,765 hospitalised patients aged 20-75, who required supplemental oxygen in the first 3 days in hospital**

| Measure | Distribution/ Event rate | Estimated treat- ment effect | Total n (β = 80% 2p = 0.05) |
|---|---|---|---|
| Opportunistic $S/F_{94}$ day 5 | Mean = 2.39 SD = 1.29 $\rho$ vs Day 0 = 0.31 | $\Delta S/F_{94}$: 0.18 | 1444 |
| Protocolised $S/F_{94}$ day 5 | Mean = 2.39 SD = 1.25 $\rho$ vs Day 0 = 0.57 | $\Delta S/F_{94}$: 0.18 | 988 |
| WHO day 5 | (See Supplementary Table 4) | OR: 0.84 | 3331 |
| 1-level sustained improvement | 13,437/ 30,060 (44.7%) | RR: 1.03 | 6756 |
| 2-level sustained improvement | 5411/ 30,060 (18.0%) | RR: 1.04 | 3808 |
| 28-day mortality | 8262/39,765 | RR: 0.85 | 5143 |

The estimated treatment effect is for a 15% relative reduction in mortality. Sample size shows the total number of subjects needed in both arms to detect the estimated treatment effect shown, using a 1:1 allocation. Protocolised $S/F_{94}$ - hypothetical improvement in power using a protocolised measurement of $S/F_{94}$. $\Delta S/F_{94}$ - change in $S/F_{94}$ associated with a 15% reduction in mortality. *RR* risk ratio. *OR* proportional odds ratio.

reduction in mortality based on previous power calculations for the RECOVERY trial. We then estimated the sample sizes required to detect these effects with 80% power at $2p = 0.05$ (2p indicates a two-tailed test).

Some examples of sample size estimations using different inclusion criteria can be found in the supplementary material (Supplementary Table 2 and Supplementary Table 3). We created an online tool, using synthetic data with similar characteristics to the ISARIC4C data (see Methods), to enable users to test any combination of inclusion criteria (age, frailty score and type of respiratory support) and outcome assessment timepoint: https://isaric4c.net/endpoints.

For a 15% relative reduction in mortality, the required sample size was smallest for $S/F_{94}$ on day 5, needing 722 patients in each arm (1444 in total, Table 1). The number of subjects required for $S/F_{94}$ on day 8 was higher, with 1,342 subjects in each arm (Supplementary Table 4). For the WHO ordinal scale, 1,666 participants would be required in each arm on day 5, or 1,168 on day 8 to detect this mortality reduction. Required sample size was larger when 1-level sustained improvement was used as the outcome variable, with 3,378 patients in each arm, and 1,904 subjects in each arm when using 2-level sustained improvement (Table 1). Errors around the point estimates shown in Table 1 are shown in Fig. 3 for a range of effect sizes.

**Estimated improvement with protocolised measurement of $S/F_{94}$**

We have developed a protocol for measurement of $S/F_{94}$ (Appendix: Protocol). Protocolising measurements is likely to substantially improve the accuracy of measurements of oxygenation function, firstly by ensuring that an oxygen delivery mode is used for which $F_IO_2$ can be accurately quantified (e.g. Venturi systems), and secondly by ensuring that measurements are taken at steady state. Protocolised measurement also permits inclusion of all patients, since $F_IO_2$ is decreased until $S_pO_2 \leq 0.94$, to a minimum of $F_IO_2 = 0.21$. We sought to estimate the magnitude of this improvement. We did this by fitting a measurement error model relating opportunistic and protocolised $S/F_{94}$ measurements. A description of the estimation of effect size for the protocolised $S/F_{94}$ measurement can be found in the supplementary methods. Based on this effect size estimate, the required sample size for a protocolised measurement of day 5 $S/F_{94}$ would be around 988 subjects in total (Fig. 3).

## Discussion

In synthetic (Fig. 1) and real (Supplementary Fig. 1) physiological data, we found that $S_aO_2 \leq 0.94$ is a pragmatic cut-off threshold, lying within a safe range, excluding the majority of obviously misleading values caused by the ceiling effect, and optimising predictive validity. Using observational data from the ISARIC4C study, we demonstrate that $S/F_{94}$ fulfills our initial requirements for an intermediate outcome: a continuous outcome measure that is closely related to mortality and can be modified by therapy[3]. Testing predicted statistical power for a range of effect sizes in observational data, we found that $S/F_{94}$ is more sensitive than other widely-used outcomes. Comparing both the WHO ordinal scale and $S/F_{94}$ to the definitive outcome of mortality at day 28, we found that the same predicted treatment effect can be detected with fewer patients using $S/F_{94}$, even when measurements are not protocolised.

In a clinical trial setting, where both $S_pO_2$ and $F_IO_2$ measurement can be protocolised, sensitivity is predicted to improve because protocolised measurement are less noisy and are therefore expected to have a stronger relationship with mortality. Using the SD for protocolised $S/F_{94}$ during the RECOVERY trial, together with the assumed error measurement model relating protocolised and opportunistic $S/F_{94}$ measurements, we predict a substantial additional improvement in statistical power using a protocolised measurement.

Our analyses may underestimate the statistical power of mortality, since time-to-event analyses would be used in most circumstances to maximise statistical power. Due to the large proportion of missing data after day 10, it was not possible to carry out survival modelling in our data. Ideally, we would have performed a mediation analysis with treatment effect, to determine the extent to which the treatment effect on mortality is explained by the intermediate endpoint $S/F_{94}$. However, since there is no $S/F_{94}$ data available from clinical studies showing significant treatment effect, it is not possible to perform this analysis.

Some important sources of error exist in the outcome measures we considered. Firstly, $S_pO_2$ and $F_IO_2$ are both subject to measurement error, particularly in opportunistic data. For example, estimating $F_IO_2$ for patients receiving supplemental oxygen via nasal cannula or simple (Hudson) masks is inaccurate, because the $F_IO_2$ is profoundly affected by inspiratory flow rate, which varies between patients. This error would be eliminated by protocolised measurement, which mandates the use of devices delivering a fixed $F_IO_2$. Secondly, the position of a patient on the ordinal WHO scale is influenced by both availability of resources and the decision by the patient and the clinician whether to escalate the level of care or provide organ support. This may explain the wide range of $S/F_{94}$ values for patients at the same position on the WHO scale.

There are multiple advantages of using $S/F_{94}$ as an intermediate outcome measure in a phase II clinical trial in hospitalised patients. It is an easy, non-invasive measurement, using near-ubiquitous monitoring equipment. In contrast, daily $P_aO_2$ measurements (from an arterial blood sample) are time-consuming, require highly skilled staff, and are burdensome for patients unless an indwelling arterial catheter is present (unusual outside of critical care areas). It is likely that the results of recent and ongoing clinical trials suggesting harm from hyperoxia will, in future, mean high $S_aO_2$ values a less common finding, particularly in the intensive care unit.

In order to determine the utility of a surrogate outcome in clinical trials, a distinction can be made between "individual level surrogacy" and "trial-level surrogacy"[18]. If there is an association between the surrogate and the outcome of interest in individual patients, the surrogate works on an individual level. If the effect that a treatment has on the surrogate can be used to predict the causal effect treatment has on the outcome, there is also trial level surrogacy. There are some scenarios, as explained by Buyse and colleagues[18], in which there is individual-level surrogacy but no trial level surrogacy, for example due

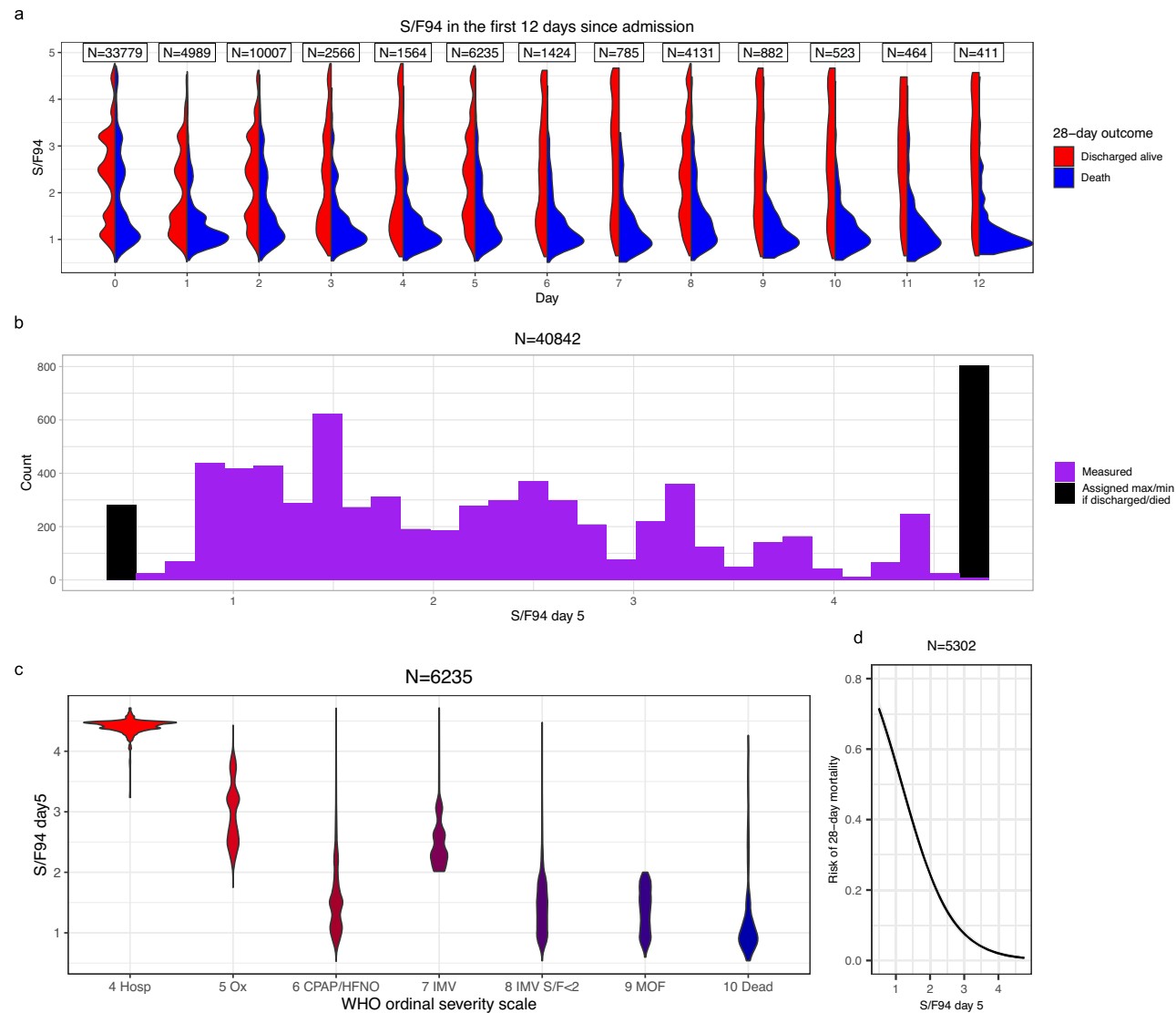

**Fig. 2 | Evaluation of $S/F_{94}$ in observational data. a** Smoothed distributions of $S/F_{94}$ values in survivors and non-survivors during the first 12 days of the study, not including assigned minimum/maximum values (restricted to 39,765 patients aged 20 − 75, oxygen therapy within 3 days). **b** Histogram showing distribution of $S/F_{94}$ values on day 5 as used for subsequent analyses (in purple). Patients discharged home before day 5 are assigned the maximum value (4.78), and patients who died before day 5 are assigned to an arbitrary minimum of 0.5 (in black). **c** Distribution of $S/F_{94}$ values day 5 compared with WHO ordinal scale[2] value at the same time point, in patients who met our inclusion criteria (aged 20 − 75, oxygen therapy within

3 days). No assigned minimum or maximum values are included in this figure. Hosp = hospitalised, no oxygen support; Ox = Hospitalised, oxygen by mask or nasal prongs; CPAP/HFNO = Hospitalised, oxygen by continuous positive airway pressure; high-flow nasal oxygen or non-invasive ventilation; IMV = Intubation and mechanical ventilation; IMV $S/F ≤ 2$ = Mechanical ventilation; $S/F ≤ 2$ or vaso-pressors; MOF = Multi-organ failure & mechanical ventilation & $S/F ≤ 2$ & ECMO or renal replacement therapy. **d** Logistic regression analysis with 95% confidence interval, using both $S/F_{94}$ on day 0 and $S/F_{94}$ on day 5 as covariates, showing a clear association between mortality at 28 days and $S/F_{94}$ value on day 5.

to (known and unknown) confounders, or treatment being dependent on the surrogate (e.g. low $S/F_{94}$ values could lead to additional inter-ventions that influence the outcome, confounding the influence of treatment on outcome). Trial-level surrogacy can be demonstrated with data from (multiple) randomised controlled trials. With the data we have available, we can thus only show individual-level surrogacy and not trial-level surrogacy. Determining whether $S/F_{94}$ is also a trial-level surrogate would be a desirable objective for further studies.

Of the pragmatic endpoints available from routinely collected data, the WHO ordinal scale is the best-performing endpoint. In studies where clinical observations can be obtained, $S/F_{94}$ is a robust measure of pulmonary oxygenation function, and is the best measure to optimise statistical power for comparisons. $S/F_{94}$ is comparable to the P/F ratio as a measure of pulmonary oxygenation, and superior to $S_pO_2/F_IO_2$ ratio. Where protocolised measurements can be obtained, further

improvements in statistical power are expected. $S/F_{94}$ may have utility in clinical studies of other disease processes where pulmonary oxgena-tion failure contributes to mortality, such as influenza and ARDS[19].

In conclusion, $S/F_{94}$ is a powerful and robust intermediate end-point for clinical studies of COVID-19 and may have broad utility in forms of acute lung injury.

## Methods

### Ethical approval

All research described in this study complies with all relevant ethical regulations. Ethical approval was given by the South Central-Oxford C Research Ethics Committee in England (13/SC/0149), the Scotland A Research Ethics Committee (20/SS/0028), and the WHO Ethics Review Committee (RPC571 and RPC572, April 2013). In England and Wales, consent was not required for the collection of depersonalised routine

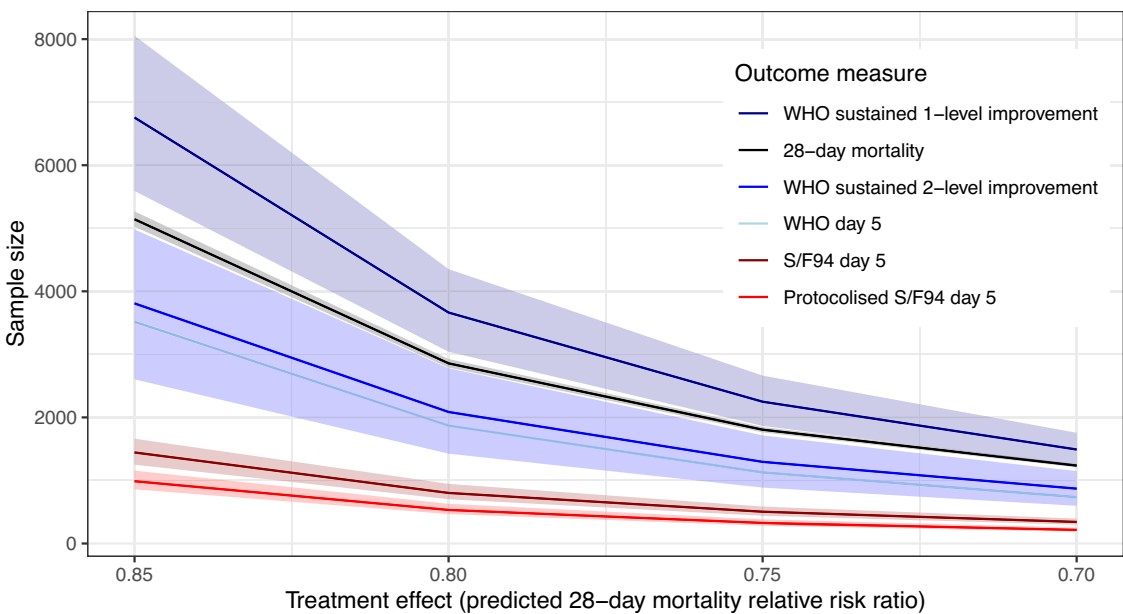

**Fig. 3 | Comparison of the number of patients needed, including 95% confidence interval, for the different outcome measures, using treatment effects between 0.85 and 0.70.** The bottom line shows predicted sample size required when using a protocolised $S/F_{94}$ measurement, rather than an opportunistic measurement.

healthcare research data. In Scotland, a waiver for consent was given by the Public Benefit and Privacy Panel.

### Relationship to the reference standard ($P/F$ ratio)

The $P/F$ ratio is the oxygenation measure used in diagnostic criteria for acute respiratory failure, and is used in our analysis as the reference standard[20]. We evaluated the relationship between $S/F$ and $P/F$ in two datasets: a synthetic dataset of 1,529,176 predictions covering a wide range of possible physiological variation, generated by a mathematical model of oxygen delivery written in Python (available at https://github.com/baillielab/oxygen_delivery) and reported previously[17], and 72,457 unselected arterial blood gas results from a critically ill population[17]. Taking $P/F$ to be our reference standard, we evaluated $S/F$ at different thresholds in both synthetic and real data.

### Predictive validity

We considered the predictive validity of $S/F$ and $S/F_{94}$ compared to $P/F$ and two other measures of oxygenation function: the A-a, and effective shunt fraction (ES)[17].

Predictive validity quantifies the extent to which a clinical measurement predicts an unseen event. The aim is not to optimise prediction, but to test the extent to which a measurement is describing a real feature of the patient's illness[21]. In this case, we contend that a measure that accurately describes pulmonary oxygenation function will accurately predict $P_aO_2$ after a change is made to $F_IO_2$. Using the same pre-existing dataset of ABG results from critically ill patients as in our previous study[17], we used this approach to assess the validity of $S/F$ and $S/F_{94}$.

Briefly, in pairs of arterial blood gas results taken from the same patient <3 h apart, in which $F_IO_2$ was decreased in the later sample, we used various measures of oxygenation (A-a, $P/F$, ES, $S/F$) in the first ABG to predict the $P_aO_2$ in the second sample and compared these predicted values with the $P_aO_2$ that was measured in the second sample. Predictive validity was quantified by the median absolute error (MAE). A baseline value, showing the difference between ABG results for matched pairs in which $F_IO_2$ did not change, is provided to contextualise the MAE results as a reasonable minimum error value. Results are presented as difference in MAE from this baseline. The

Mann-Whitney U-test (MWU) was used for the comparison of MAE difference from baseline.

### Evaluation in ISARIC4C data

**Inclusion criteria.** All subjects were part of the ISARIC Coronavirus Clinical Characterisation Consortium (ISARIC4C) WHO Clinical Characterisation Protocol UK (CCP- UK), a study in England, Wales, and Scotland prospectively collecting data from patients hospitalised with SARS-CoV-2 infection since the start of the pandemic.

In order to focus our assessment on the subset of patients with hypoxaemic respiratory failure that is potentially modifiable by anti-inflammatory treatment, we repeated all analyses in subjects aged 20-75 who required supplementary oxygen therapy within 3 days of hospital admission, subjects aged 20-75 that were oxygen dependent on the day of admission, and subjects aged 20-75 without criteria for oxygen dependency. All included patients had $S_pO_2$ and $F_IO_2$ data available. While $S_pO_2$ is typically represented as a percentage, for $S/F_{94}$ it is used as a fraction, with values ranging from 0-1.

**Estimation of $S/F_{94}$ in observational data.** The $S/F$ ratio was calculated by dividing $S_pO_2$ by $F_IO_2$ (with both as fractions, taking values between 0 and 1). For this evaluation, $S/F_{94}$ was defined as an opportunistic measurement in which $S_pO_2 \leq 0.94$, or the patient was receiving no supplementary oxygen ($F_IO_2 = 0.21$).

Importantly, the retrospectively-defined subgroup of patients meeting the $S/F_{94}$ criteria is not representative of all patients since there was an excess of patients who were not receiving respiratory support, with slight excess mortality, in the $S/F_{94}$ group (Supplementary Table 1). This indicates at least two mechanisms of selection bias, acting in opposite directions, and precluding a direct comparison. Firstly, patients who have high blood oxygen levels on relatively little supplementary oxygen are excluded from the $S/F_{94}$ group; by definition these patients have relatively mild disease. Secondly, the group in whom $S/F_{94}$ could be measured includes patients who receive supplemental oxygen, and fail to reach adequate $S_pO_2$ values, but are not escalated to a higher level of respiratory support; this is a frail and multimorbid population with very severe disease.

$S/F_{94}$ was calculated at baseline (day 0) and on day 5 and day 8 from study enrolment. There is expected to be differential missingness between $S/F_{94}$ and mortality: $S_pO_2$ and $F_IO_2$ data are only available for a proportion of cases, whereas outcome data is well-recorded. Patients who died or were discharged on given day and had a missing value for $S/F_{94}$ were assigned values 0.5 (severe oxygenation defect) and 4.76 (perfect oxygenation), respectively. However, death/discharge was more likely to be recorded than $S/F_{94}$, and this could introduce bias into our analysis. We addressed this by estimating the proportion of patients for whom $S/F_{94}$ measurements were available among those who had not died or been sent home by a given day. We then resampled those who died/discharged according to these proportions. For example, if on day 5, 20% of those who had not died or discharged had $S/F_{94}$ measurements available, we randomly resampled 20% of those who died/had been discharged by then, assigning $S/F_{94} = 0.5$ to those who died, and $S/F_{94} = 4.76$ to those who were discharged.

**Association between $S/F_{94}$ and 28-day mortality.** Two key assumptions underlie the use of $S/F_{94}$ as an intermediate endpoint. Firstly, that pulmonary oxygenation function predicts mortality in COVID-19, and secondly, that $S/F_{94}$ accurately reflects the pulmonary oxygenation function. If either of these assumptions are violated, then a strong relationship between $S/F_{94}$ and subsequent mortality would not be expected.

To evaluate this association, a logistic regression model was developed with 28-day all-cause mortality as the dependent variable and $S/F_{94}$ measured on day 0 and day 5 as two separate covariates. We included both $S/F_{94}$ on day 0 and day 5 due to the strong relationship between $S/F_{94}$ on day 0 and $S/F_{94}$ on days further in the disease trajectory. Linear dependence of log-odds on $S/F_{94}$ measured on day 0 and day 5 was assessed both by visual inspection and using model selection criteria including the Bayesian Information Criterion (BIC) to compare to a restricted splines model. Finally, predicted models were made to assess the absolute change in risk of mortality with a change in $S/F_{94}$.

**Sample size calculations.** We compared the sample sizes required for a range of different outcomes measures ($S/F_{94}$, WHO ordinal scale, sustained improvement at day 28 and 28-day mortality). For the intermediate endpoints, we estimated the treatment effect associated with a 15% relative reduction in mortality. Below we give brief descriptions of the effect size calculations for the different outcome measures. All calculations assumed a 1:1 allocation of participants between treatment and control groups and are based on having 80% power at $2p = 0.05$ to detect the stated treatment effect. Details on effect size estimation can be found in the supplementary material.

**Quantifying uncertainty.** We bootstrapped 95% confidence for the effect size, and then used this to calculate 95% confidence intervals for required sample size using the fact that they are monotonically related.

**Continuous variables ($S/F_{94}$).** We fit a logistic regression with mortality at day 28 as the dependent variable, and age, sex, $S/F_{94}$ on day 0 (baseline) and day 5 (or day 8) as independent variables. We used this to calculate the predicted probability of mortality, and the change in $S/F_{94}$ associated with a relative reduction in predicted mortality of 15%, for each subject. Finally, we took the mean to find the average change in day 5 $S/F_{94}$ that is associated with a 15% reduction in mortality across the sample. This was the target treatment effect in the clinical trial. We calculated the sample size required to see this treatment effect with a given level of power using a two sample t-test with ANCOVA correction for the correlation between $S/F_{94}$ on day 0 and day 5[22].

**Ordinal variables (WHO scale).** Values for the WHO ordinal scale were derived using information about oxygen support and mortality. Possible values in hospitalised patients range between 4 and 10[2].

**WHO scale - absolute value.** We fitted a proportional odds model with the WHO ordinal scale as the dependent variable, and age and sex as independent variables. We used this model to estimate the odds ratio associated with a 15% relative reduction in mortality[23].

**WHO scale - sustained improvement.** We derived binary variables for sustained 1- or 2-level improvement on the WHO scale. To be considered sustained, an improvement had to be maintained until discharge or until day 28. We fitted a logistic regression model with mortality at day 28 as the dependent variable, and age, sex and sustained 1- or 2-level improvement on the WHO scale as independent variables. We used this model to estimate the difference in proportion of people who had a sustained improvement on the WHO ordinal scale that was associated with a 15% reduction in risk of mortality. We then calculated required sample size for this outcome using a two-sample test for proportions with a continuity correction[24]. Only patients who had WHO ordinal scale values on at least two separate days were included in this analysis.

**Mortality.** In order to compare these alternative outcome measures with a definitive outcome (mortality), we calculated the number of participants needed if 28-day mortality was the outcome measure, using a two-sample test for proportions with continuity correction.

### Reporting summary
Further information on research design is available in the Nature Portfolio Reporting Summary linked to this article.

### Data availability
Source data are provided for Fig. 1 and supplementary figure 1 and 2. The dataset used and analysed in this study contains clinical data about individuals and is available after a data access request. Data access request and details on the procedure can be found at https://odap.ac.uk/researchers. Data access requests will be reviewed on the basis of scientific merit and validity, the proposed timeline, ethical considerations and the available resources. Access requests can be send to odap@ed.ac.uk. A reply to a data access request will be provided within six weeks from the date of the request. Depending on the requested data, there may be additional steps before data can be published, such as agreement from all contributors. For details, please see https://odap.ac.uk/researchers. All data supporting the findings in this manuscript are present in the main text, supplementary material, the source data and from the corresponding author upon request. A synthetically generated dataset, containing the same key properties as the original dataset is available for sample size calculations on https://isaric4c.net/endpoints Source data are provided with this paper.

### Code availability
The code used to do the analyses can be found on github https://github.com/baillielab/SF94.

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

## Acknowledgements

This work uses data provided by patients and collected by the NHS as part of their care and support. We are extremely grateful for the front-line NHS clinical and research staff and volunteer medical students who collected this data in challenging circumstances, and the generosity of the participants and their families for their individual contributions in these difficult times. We also acknowledge the support of Jeremy J Farrar (Wellcome Trust) and Nahoko Shindo (WHO). For the purpose of open access, the author has applied a CC BY public copyright licence to any Author Accepted Manuscript version arising from this submission. JKB gratefully acknowledges funding support from a Wellcome Trust Senior Research Fellowship (223164/Z/21/Z), UKRI grants MC_PC_20004, MC_PC_19025, MC_PC_1905, MRNO2995X/1, and MC_PC_20029, Sepsis Research (Fiona Elizabeth Agnew Trust), a BBSRC Institute Strategic Programme Grant to the Roslin Institute (BB/P013732/1, BB/P013759/1), and the UK Intensive Care Society. ISARIC4C work was supported by the National Institute for Health Research (NIHR), the Medical Research Council [MC_PC_19059] and by the NIHR Health Protection Research Unit (HPRU) in Emerging and Zoonotic Infections at University of Liverpool in partnership with Public Health England (PHE), in collaboration with Liverpool School of Tropical Medicine and the University of Oxford [200907], NIHR HPRU in Respiratory Infections at Imperial College London with PHE [200927], Wellcome Trust and Department for International Development [215091/Z/18/Z], and the Bill and Melinda Gates Foundation [OPP1209135], and Liverpool Experimental Cancer Medicine Centre (C18616/A25153), NIHR Biomedical Research Centre at Imperial College London [IS-BRC-1215-20013], EU Platform for European Preparedness Against (Re-) emerging Epidemics (PREPARE) [FP7 project 602525] and NIHR Clinical Research Network for providing infrastructure support for this research.

## Author contributions

J.K.B. and P.H. conceived the study. J.K.B., M.G.S. and P.J.M.O. acquired funding. J.K.B., P.W.H., F.M., N.Y., J.D., A.D.B., J.M., J.S.N.-V.-T., P.W.H. and M.G.S. designed the analysis. E.M.H., R.G., E.S., A.B.D., D.H., K.R., N.S. and N.L. provided guidance on methodology and interpretation. M.C.S., S.K., A.B.B., N.S. and J.K.B. did the formal analysis. J.S.B. and S.K. created the website. E.H., A.B.D., G.H.G., N.L., N.S. and J.K.B. supervised the work. M.C.S., S.K. and J.K.B. wrote the original draft of the manuscript. All authors reviewed and gave feedback on the manuscript. All authors read and approved the final manuscript.

## Competing interests

JKB and ABD report grants from the UK Department of Health and Social Care (DHSC), during the conduct of the study, and grants from Wellcome Trust,. PJMO reports personal fees from consultancies (GlaxoSmithKline, Janssen, Bavarian Nordic, Pfizer, and Cepheid) and from the European Respiratory Society, grants from MRC, MRC Global Challenge Research Fund, the EU, NIHR BRC, MRC–GlaxoSmithKline, Wellcome Trust, NIHR (HPRU in Respiratory Infection), and is an NIHR senior investigator outside the submitted work. PJMO's role as President of the British Society for Immunology was unpaid but travel and accommodation at some meetings was provided by the Society. JKB reports grants from MRC. MGS reports grants from DHSC, NIHR UK, MRC, HPRU in Emerging and Zoonotic Infections, and University of Liverpool, during the conduct of the study, and is chair of the scientific advisory board and a minority shareholder at Integrum Scientific, outside the submitted work. JSN-V-T was seconded to the Department of Health and Social Care, England (DHSC), October 2017-March 2022. The views expressed in this manuscript are those of its authors and not necessarily those of DHSC. JSN-V-T reports personal fees and travel and accommodation from AstraZeneca. NS reports grants from Boehringer Ingleheim and Novo Nordisk outside the submitted work. The remaining authors declare no competing interests.

## Additional information

## ISARIC4C Investigators

**Co-Investigators** J. Kenneth Baillie [1,16,17,18], Peter Openshaw [13], Malcolm G. Semple [14,15], Beatrice Alex[4], Petros Andrikopoulos[19,20], Benjamin Bach[4], Wendy S. Barclay[21], Debby Bogaert[22], Meera Chand[23], Kanta Chechi[19,24], Graham S. Cooke[25], Ana da Silva Filipe[26], Thushan de Silva[27], Annemarie Docherty [3], Gonçalo dos Santos Correia[28,29], Marc-Emmanuel Dumas[19,20,30,31], Jake Dunning[11], Tom Fletcher[32], Christopher A. Green[33], William Greenhalf[34], Julian Griffin[19], Rishi Gupta [5], Ewen M. Harrison[3], Antonia Y. W. Ho[26,35], Karl Holden[36], Peter W. Horby [11], Samreen Ijaz[37], Say Khoo[38], Paul Klenerman[39,40], Andrew Law[1], Matthew Lewis[28,29], Sonia Liggi[19], Wei Shen Lim[41], Lynn Maslen[28,29], Alexander J. Mentzer[42,43], Laura Merson[44], Alison M. Meynert[8], Shona C. Moore[45], Mahdad Noursadeghi[46], Michael Olanipekun[19,20], Anthonia Osagie[19,20], Massimo Palmarini[26], Carlo Palmieri[47,48], William A. Paxton[45,49], Georgios Pollakis[45,49], Nicholas Price[50,51], Andrew Rambaut[52], David L. Robertson[26], Clark D. Russell[22], Vanessa Sancho-Shimizu[53], Caroline Sands[28,29], Janet T. Scott[26,54], Louise Sigfrid[44], Tom Solomon[55,56], Shiranee Sriskandan[25,57], David Stuart[58], Charlotte Summers[59], Olivia V. Swann[60], Zoltan Takats[19,61], Panteleimon Takis[28,29], Richard S. Tedder[62,63,64], A. A. Roger Thompson[65], Emma C. Thomson[26], Ryan S. Thwaites[13], Lance C. W. Turtle[55,66], Maria Zambon[67] & Gail Carson[68]

**Data analysis team** Thomas M. Drake[3], Cameron J. Fairfield[3], Stephen R. Knight[3], Kenneth A. Mclean[3], Derek Murphy[3], Lisa Norman[3], Riinu Pius[3] & Catherine A. Shaw[3]

**Data architecture team** Marie Connor[68], Jo Dalton[68], Carrol Gamble[68], Michelle Girvan[68], Sophie Halpin[68], Janet Harrison[68], Clare Jackson[68], Laura Marsh[68], Stephanie Roberts[68], Egle Saviciute[68], Sara Clohisey[1], Ross Hendry[1], Susan Knight[69], Eva Lahnsteiner[69], Andrew Law[1], Gary Leeming[70], Lucy Norris[71], James Scott-Brown [4], Sarah Tait[69] & Murray Wham[8]

**Data analysis and management team** James Lee[44], Daniel Plotkin[44] & Seán Keating[16]

**Project administration team** Cara Donegan[14] & Rebecca G. Spencer[14]

**Project management team** Chloe Donohue[68], Fiona Griffiths[1], Hayley Hardwick[55] & Wilna Oosthuyzen[1]

[19]Section of Biomolecular Medicine, Division of Systems Medicine, Department of Metabolism, Digestion and Reproduction, London, UK. [20]Section of Genomic and Environmental Medicine, Respiratory Division, National Heart and Lung Institute, London, UK. [21]Section of Molecular Virology, Imperial College London, London, UK. [22]Centre for Inflammation Research, The Queen's Medical Research Institute, University of Edinburgh, Edinburgh, UK. [23]Antimicrobial Resistance and Hospital Acquired Infection Department, Public Health England, London, UK. [24]Department of Epidemiology and Biostatistics, School of Public Health, Faculty of Medicine, Imperial College London, London, UK. [25]Department of Infectious Disease, Imperial College London, London, UK. [26]MRC-University of Glasgow Centre for Virus Research, Glasgow, UK. [27]The Florey Institute for Host-Pathogen Interactions, Department of Infection, Immunity and Cardiovascular Disease, University of Sheffield, Sheffield, UK. [28]National Phenome Centre, Department of Metabolism, Digestion and Reproduction, Imperial College London, London, UK. [29]Section of Bioanalytical Chemistry, Department of Metabolism, Digestion and Reproduction, Imperial College London,

London, UK. [30]European Genomic Institute for Diabetes, CNRS UMR 8199, INSERM UMR 1283, Institut Pasteur de Lille, Lille University Hospital, University of Lille, Lille, France. [31]McGill University and Genome Quebec Innovation Centre, Montréal, QC, Canada. [32]Liverpool School of Tropical Medicine, Liverpool, UK. [33]Institute of Microbiology and Infection, University of Birmingham, Birmingham, UK. [34]Department of Molecular and Clinical Cancer Medicine, University of Liverpool, Liverpool, UK. [35]Department of Infectious Diseases, Queen Elizabeth University Hospital, Glasgow, UK. [36]University of Liverpool, Liverpool, UK. [37]Virology Reference Department, National Infection Service, Public Health England, Colindale Avenue, London, UK. [38]Department of Pharmacology, University of Liverpool, Liverpool, UK. [39]Nuffield Department of Medicine, Peter Medawar Building for Pathogen Research, University of Oxford, Oxford, UK. [40]Translational Gastroenterology Unit, Nuffield Department of Medicine, University of Oxford, Oxford, UK. [41]Nottingham University Hospitals NHS Trust:, Nottingham, UK. [42]Nuffield Department of Medicine, John Radcliffe Hospital, Oxford, UK. [43]Department of Microbiology/Infectious Diseases, Oxford University Hospitals NHS Foundation Trust, John Radcliffe Hospital, Oxford, UK. [44]ISARIC Global Support Centre, Centre for Tropical Medicine and Global Health, Nuffield Department of Medicine, University of Oxford, Oxford, UK. [45]Institute of Infection, Veterinary and Ecological Sciences, University of Liverpool, Liverpool, UK. [46]Division of Infection and Immunity, University College London, London, UK. [47]Molecular and Clinical Cancer Medicine, Institute of Systems, Molecular and Integrative Biology, University of Liverpool, Liverpool, UK. [48]Clatterbridge Cancer Centre NHS Foundation Trust, Liverpool, UK. [49]NIHR Health Protection Research Unit in Emerging and Zoonotic Infections, Liverpool, UK. [50]Centre for Clinical Infection and Diagnostics Research, Department of Infectious Diseases, School of Immunology and Microbial Sciences, King's College London, London, UK. [51]Department of Infectious Diseases, Guy's and St Thomas' NHS Foundation Trust, London, UK. [52]Institute of Evolutionary Biology, University of Edinburgh, Edinburgh, UK. [53]Department of Pediatrics and Virology, St Mary's Medical School Bldg, Imperial College London, London, UK. [54]NHS Greater Glasgow & Clyde, Glasgow, UK. [55]NIHR Health Protection Research Unit for Emerging and Zoonotic Infections, Institute of Infection, Veterinary and Ecological Sciences University of Liverpool, Liverpool, UK. [56]Walton Centre NHS Foundation Trust, Liverpool, UK. [57]MRC Centre for Molecular Bacteriology and Infection, Imperial College London, London, UK. [58]Wellcome Centre for Human Genetics, University of Oxford, Roosevelt Drive, Oxford, UK. [59]Department of Medicine, University of Cambridge, Cambridge, Cambridgeshire, UK. [60]Department of Child Life and Health, University of Edinburgh, Edinburgh, UK. [61]National Phenome Centre, Division of Systems Medicine, Department of Metabolism, Digestion and Reproduction, Imperial College London, London, UK. [62]Blood Borne Virus Unit, Virus Reference Department, National Infection Service, Public Health England, London, UK. [63]Transfusion Microbiology, National Health Service Blood and Transplant, London, UK. [64]Department of Medicine, Imperial College London, London, UK. [65]Department of Infection, Immunity and Cardiovascular Disease, University of Sheffield, Sheffield, UK. [66]Tropical & Infectious Disease Unit, Royal Liverpool University Hospital, Liverpool, UK. [67]National Infection Service, Public Health England, London, UK. [68]Liverpool Clinical Trials Centre, University of Liverpool, Liverpool, UK. [69]Public Health Scotland, Scotland, UK. [70]Centre for Health Informatics, Division of Informatics, Imaging and Data Science, School of Health Sciences, Faculty of Biology, Medicine and Health, University of Manchester, Manchester Academic Health Science Centre, Manchester, UK. [71]EPCC, University of Edinburgh, Edinburgh, UK.

