## [Peer Review File · Nature Communications]

Evaluation of pragmatic oxygenation measurement as a proxy for Covid-19 severityEditorial Note: This manuscript has been previously reviewed at another journal that is not operating a transparent peer review scheme. This document only contains reviewer comments and rebuttal letters for versions considered at *Nature Communications*.

REVIEWERS' COMMENTS

Reviewer #3 (Remarks to the Author):

The authors have addressed my comments well.

Reviewer #5 (Remarks to the Author):

I think that the authors satisfactorily answered to reviewer 1 critiques. I have 3 minor comments that could be taken into account:

- The predictive validity is not clear to me: I think that the authors mean that SF can predict PaO₂ in an abg taken at the same time rather than in the future. The correlation is at the same timepoint but this isn't clear from what is written
- It's not clear on what day the prediction of mortality from SF94 is maximal (fig2a), is there any threshold, too?
- Finally the SF94 name is not so clear to me, as patients could have values higher than 94 at FiO₂ 0.21 and still included (see the misunderstanding on point 5. by reviewer 1). What about something like SF_{dyn} for dynamic (you have to decrease FiO₂ to 94% SpO₂ or 0.21 FiO₂) or SF_{lin} for linear (avoiding the non linear ceiling effect?)

Reviewer #6 (Remarks to the Author):

The authors have done a nice job responding to the comments. I have no further comments.

Reviewer #7 (Remarks to the Author):

The authors developed a pragmatic oxygenation measurement as a proxy for Covid-19 severity. Using large databases from the RECOVERY trial and the ISARIC4C study, they proposed to consider the SaO₂ levels only up to 94% (what they call *S/F94*) to evaluate the pulmonary oxygenation function. We know the limitation of pulse oximetry measurements in the high PaO₂ range, as it is depicted by the oxyhemoglobin dissociation curve.

There is a discrepancy between the impressive work produced here and the limited novelty of the findings.

Nevertheless, the study is well done.

Major comments:

1- An important question is the necessity to exclude SpO₂ values above 94% if we decide to avoid such higher SpO₂ values in current practice. The importance of avoiding high SpO₂ is now established to avoid hyperoxaemia, as this could be harmful.

To cite some examples, van den Boom et al (Chest 2020) found from the MIMIC III data base that a 94-98% range may be optimal. In 2017, Vincent (Lancet) proposed to keep SpO₂ between 92 and 96%, and the British Thoracic Society Guideline (BMJ 2017) considered that a slightly lower range of 90-94% may be adequate. Maybe the present findings will help to modify our practices (and the design of clinical trials)?. This should be discussed.

2- The relation between the severity of alterations in gas exchange and outcome is well known in COVID-19 like in other diseases. These results can be summarized, but the discussion section should include a short comment about this.

Additional comments:

1- The authors should briefly discuss whether the relation between SpO₂ and PaO₂ may be different in COVID-19 than in other settings

2- The authors should briefly discuss the harmful effects of hyperoxaemia (undetected when SpO₂ is close to 100%).

Jean-Louis Vincent

response to reviewers- S/F_{94}

We would like to thank the reviewers for taking the time to read our revised manuscript and the suggestions for further improvement.

Reviewer #3: Remarks to the Author

The authors have addressed my comments well.

Reviewer #5: Remarks to the Author

I think that the authors satisfactorily answered to reviewer 1 critiques. I have 3 minor comments that could be taken into account:

1. The predictive validity is not clear to me: I think that the authors mean that SF can predict PaO₂ in an abg taken at the same time rather than in the future. The correlation is at the same timepoint but this isn't clear from what is written

Predictive validity refers to the ability of a test to predict a future, unseen outcome. We have done this using a data on sets of arterial blood gas (ABG) samples, that were taken within a 3 hour timeframe. The blood samples were taken from patients in whom underlying pulmonary pathology was expected to be stable between the two samples, but the $F_I O_2$ was decreased. A smaller error (mean absolute error, MAE) between the predicted and the actual measurement indicates a better predictive validity. We have edited the manuscript, and the text now reads as follows:

“Briefly, in pairs of arterial blood gas results taken from the same patient <3h apart, in which $F_I O_2$ was decreased in the later sample, we used various measures of oxygenation (A-a, P/F , ES, S/F) in the first ABG to predict the $P_a O_2$ in the second sample **and compared these predicted values with the $P_a O_2$ that was measured in the second sample.**”

2. It's not clear on what day the prediction of mortality from SF94 is maximal (fig2a), is there any threshold, too?

We did not calculate a threshold for the prediction of mortality from S/F_{94} , but there is a clear increase in predictive validity over time (as is expected for a progressive disease like COVID-19). The decision to use day 5 as the endpoint in our study was based on a combination of data availability and plausibility.

3. Finally the SF94 name is not so clear to me, as patients could have values higher than 94 at FiO2 0.21 and still included (see the misunderstanding on point 5. by reviewer 1). What about something like SFdyn for dynamic (you have to decrease FiO2 to 94% SpO2 or 0.21 FiO2) or SFlin for linear (avoiding the non linear ceiling effect?)

We thank the reviewer for this suggestion and have discussed it within our team. We agree that SFlin in particular is an attractive alternative. However for two reasons we would prefer to continue with SF94: firstly, it is possible that other groups will use different thresholds to achieve the same aim. Whilst not desirable, we anticipate that this may lead to confusion if the threshold is not included in the name of the measure. Secondly, we have already used the SF94 name in the UK-wide RECOVERY study so there is an additional risk of confusion if we change the name now.

Reviewer #6: Remarks to the Author

The authors have done a nice job responding to the comments. I have no further comments.

Reviewer #7: Remarks to the Author

The authors developed a pragmatic oxygenation measurement as a proxy for Covid-19 severity. Using large databases from the RECOVERY trial and the ISARIC4C study, they proposed to consider the SaO2 levels only up to 94% (what they call S/F_{94}) to evaluate the pulmonary oxygenation function. We know the limitation of pulse oximetry measurements in the high PaO2 range, as it is depicted by the oxyhemoglobin dissociation curve. There is a discrepancy between the impressive work produced here and the limited novelty of the findings. Nevertheless, the study is well done.

Major comments:

- 1- An important question is the necessity to exclude SpO2 values above 94% if we decide to avoid such higher SpO2 values in current practice. The importance of avoiding high SpO2 is now established to avoid hyperoxaemia, as this could be harmful. To cite some examples, van den Boom et al (Chest 2020) found from the MIMIC III data base that a 94-98% range may be optimal. In 2017, Vincent (Lancet) proposed to keep SpO2 between 92 and 96%, and the British Thoracic Society Guideline (BMJ 2017) considered that a slightly lower range of 90-94% may be adequate. Maybe the present findings will help to modify our practices (and the design of clinical trials)?. This should be discussed.

We agree that this is an important area for ongoing clinical research and, in the intensive care unit particularly, we expect that future practice will make S_aO_2 values above 94% a less common clinical finding. We have amended the discussion to reflect this as suggested.

“It is likely that the results of recent and ongoing clinical trials suggesting harm from hyperoxia will, in future, mean high S_aO_2 values a less common finding, particularly in the intensive care unit.”

2- The relation between the severity of alterations in gas exchange and outcome is well known in COVID-19 like in other diseases. These results can be summarized, but the discussion section should include a short comment about this.

We agree that this is an important finding and is perhaps best demonstrated by our results in Fig 2d in the main manuscript. As the reviewer notes, this observation is well recognised in by clinicians and in the literature so we have focused our discussion on the novel findings relating to the quantification of statistical power and the implications for future research design.

Additional comments:

1- The authors should briefly discuss whether the relation between SpO₂ and PaO₂ may be different in COVID-19 than in other settings

This is an important and timely point - we have amended the discussion to clarify this and to cite the paper last week on attributable mortality in ARDS:

“ S/F_{94} may have utility in clinical studies of other disease processes where pulmonary oxygenation failure contributes to mortality, such as influenza and ARDS.¹”

2- The authors should briefly discuss the harmful effects of hyperoxaemia (undetected when SpO₂ is close to 100%).

We agree that this is an important topic and have alluded to it in an additional sentence in the discussion (see above).

References

1. Saha, R. *et al.* Estimating the attributable fraction of mortality from acute respiratory distress syndrome to inform enrichment in future randomised clinical trials. *Thorax* (2023) doi:[10.1136/thorax-2023-220262](https://doi.org/10.1136/thorax-2023-220262).